# Shielding Effectiveness and Impact Resistance of Concrete Walls Strengthened by High-Strength High-Ductility Concrete

**DOI:** 10.3390/ma14247773

**Published:** 2021-12-16

**Authors:** Jae-Hoon Lee, Jin-Seok Choi, Tian-Feng Yuan, Young-Soo Yoon

**Affiliations:** School of Civil, Environmental and Architectural Engineering, Korea University, 145 Anam-ro, Seongbuk-gu, Seoul 02841, Korea; dlwogns1994@korea.ac.kr (J.-H.L.); radiance@korea.ac.kr (J.-S.C.)

**Keywords:** electromagnetic shielding, impact resistance, strengthening, steel fiber, high-strength high-ductility concrete

## Abstract

Following the fourth Industrial Revolution, electronic and data-based technology is becoming increasingly developed. However, current research on enhancing electromagnetic interference (EMI) shielding and the physical protection performance of structures incorporating these technologies is insufficient. Therefore, in this study aiming for the improvement of EMI shielding and structural performance of structures, twelve concrete walls were fabricated and tested to determine their shielding effectiveness and drop-weight impact resistance. Concrete walls strengthened by three thickness types of high-strength, high-ductility concrete (HSDC) have been considered. The test results showed that the shielding effectiveness with strengthening thickness increased by approximately 35.6–46.2%. Specimens strengthened by more than 40% and 10% of the strengthening area ratio of single- and double-layer, respectively, exhibited more than 20 dB of shielding effectiveness. Moreover, the relationship between the damaged area ratio and shielding effectiveness was evaluated by means of the drop-weight impact test. The structural performance and EMI shielding effectiveness improved as the HSDC thickness increased.

## 1. Introduction

The fourth Industrial Revolution (Industry 4.0) has brought the importance of data-based industries to the forefront; as a result, the need for electromagnetic interference (EMI) shielding has increased to maintain the circuitry of electric devices in a normal state. EMI shielding is achieved via the loss of electric field strength when electromagnetic waves are reflected or absorbed by conductive materials. EMI shielding is generally achieved using heavy metal plates, conductive paints/spraying, or conductive mesh, which require a high initial investment and continuous maintenance, assuming that they are even effective. Few studies have been conducted on the shielding- and structural performance of undamaged structures [1,2,3,4,5,6]. Thus, it is necessary to develop widely used construction materials, such as conductive concrete, or reinforcing methods using a metal grid to ensure high shielding efficiency compared to initial outlay and evaluate the shielding effectiveness of damaged structures due to external load.

There are many types of metallic powder employed as conductive constituents, such as metal oxides, steel furnace slag, and iron. This powder needs to be incorporated into the matrix of manufactured materials at a certain volume fraction to achieve effective shielding properties; this represents the percolation threshold [1,2,3,5]. From the shielding point of view, this procedure is very inefficient due to its limited possibilities of adding materials. In recent studies, the incorporation of carbon nanotubes (CNTs) in cementitious materials exhibits some capability in terms of EMI shielding and increasing compressive and tensile strength [3,7,8]. However, it is not easy to distribute CNTs within the matrix and additional work using sonication is required. In addition, it may take some time to apply this technique to the structure due to the high price, even though a very small amount of CNTs is mixed in [9]. In contrast, metallic fibers are the most commercially available materials for construction and are very efficient for EMI shielding compared to the previously mentioned conductive materials. Fiber-reinforced concrete (FRC) also has many attractive characteristics that enhance the strength and ductility of concrete, as well as providing shielding properties [3,5]. Several researchers have commented on the use of metallic fibers in cementitious materials and their remarkable shielding effectiveness (SE), made possible through the formation of a sufficiently conductive network in the matrix [5,6,9]. The volume fraction and types of metallic fibers are considered important parameters to secure the continuity of the conduction path. Many studies [6,10,11] have been conducted on the improvement of shielding effectiveness by adding reinforcement with metallic grid and mesh, which are used as structural reinforcements instead of reinforcing bars. The grid and mesh form a regular and continuous conductive network through the narrow internal space, but their effect in shielding is remarkable only in the low-frequency band [5,6,10,11]. As the gaps in the used grids or mesh decrease, the shielding effectiveness begins to increase. However, the quality of concrete in which coarse aggregates are mixed may be reduced; as a result, only paste can be used in certain concrete covers.

It is important not only to build new buildings using the developed conductive cementitious materials or metallic grid/mesh but also to protect existing buildings from EMI. There is little research published on SE tests of existing buildings in which normal concrete has a little EMI resistance [1,2,3]. A great deal of research has been conducted to strengthen structures with fiber-reinforced cementitious materials and to perform structural evaluation [9,12]. High-strength and high-ductility concrete (HSDC) is one of the cementitious materials presenting a high level of strengthening, and high shielding performance can be expected due to the inclusion of metallic fibers.

The compressive and flexural strength properties of developed hybrid fiber-reinforced HSDC have already been evaluated (*f_ck_* > 120 MPa) and published [13,14,15]. The slant shear bond properties of interfaces between HSDC and existing normal concrete conform to the properties of the ACI Committee 546 recommendation (14–21 MPa for 28 days). The matrix of developed HSDC has low porosity and high density when manufactured through the dense packing mix design method, which uses a low water-to-binder ratio. Different lengths of polyethylene (PE) and steel fiber (SF) were used to improve the ductility, toughness, and energy-absorbing capacity of HSDC. The maximum SF volume fraction in HSDC is limited to 1.0 vol.% to secure higher tensile strain capacity and toughness through PE, affecting the strain hardening in the post-cracking zone, compared to ultra-high-performance fiber-reinforced concrete (UHPFRC), which incorporates more than 1.5 vol.% of SF. However, research in the evaluation of both EMI shielding and the impact resistance of structures strengthened by hybrid fiber-reinforced HSDC is insufficient, even though both performances are expected to be outstanding. 

Therefore, this study concentrates on research into strengthening methods to achieve high effectiveness in shielding and impact resistance through the utilization of hybrid fiber-reinforced HSDC. The variables used to evaluate the performance of EMI, impact resistance, and EMI SE after impact loading are the three different concrete thicknesses (100, 200, 300 mm), the strengthening thickness of HSDC (5, 10, 20 mm), and the layer type of HSDC (single or double). Hence, this study aims to contribute basic data regarding the EMI SE and impact resistance of reinforced concrete through fiber-reinforced cementitious composites.

## 2. Experimental Program

### 2.1. Details of Materials and Specimen Fabrication

The normal concrete (NC) mix proportion was established based on the target compressive strength of 45 MPa at 28 days, and the materials used in this paper are reported in Table 1. Type I Portland cement (density of 3.15 g/cm^3^ and specific surface area of 3413 cm^2^/g), crushed fine aggregate (density of 2.60 g/cm^3^ and fineness modulus 2.88), and coarse aggregate (density of 2.67 g/cm^3^ and fineness modulus 6.63) were used in this study, with a maximum aggregate size of 18 mm. The sieve analysis of the aggregates used is exhibited in Figure 1. The mixture proportion of HSDC is presented in Table 1. Type I Portland cement (the same as above for NC), silica fume (density 2.20 g/cm^3^ and specific surface area of 200,000 cm^2^/g), and silica filler (density 0.75g/cm^3^ and specific surface area of 2.65 cm^2^/g) were used as binder materials, as shown in Table 2. Silica sand (diameter ranging from 0.08 to 0.30 mm) was used for both fine and coarse aggregate. The high-strength straight fiber (diameter 0.2 mm and length of 19.5 mm, tensile strength of 2650 MPa and elastic modulus of 200 GPa, as shown in Table 3) and high-strength polyethylene fiber (diameter 31 μm and length of 12 mm, tensile strength of 2900 MPa and elastic modulus of 100 GPa) were adopted as hybrid fiber. The liquid polycarboxylate superplasticizer was adopted to ensure suitable workability. The results of material properties are given in Table 4, which have been established in previous research [13,14,15].

To evaluate the EMI SE and impact resistance of the reinforced concrete, twelve specimens were prepared, as detailed in Table 5. Typical square specimens, with a side length of 300 mm and thickness of 100, 200, and 300 mm, were fabricated. The naming of experimental variables is depicted in Figure 2a. The total thickness of specimens was fixed regardless of HSDC strengthening. After demolding, side surfaces of the NC specimen were given high roughness to increase the bond strength with HSDC, using metallic chiseling, then HSDC was cast onto that surface [15]. The details of strengthening with a single and double layer of HSDC are shown in Figure 2b. All test specimens were cured at a steady temperature (20 ± 1 ℃) and humidity (60 ± 5%) until the designated test day.

### 2.2. Details of Setup of Electromagnetic Interference Shielding Test

The near-field condition method was used for the EMI SE test of concrete specimens, in which customized EMI instrumentation (Jinju, Republic of Korea) utilizes a frequency range of 400–1400 MHz based on the transmitting and receiving facility. This method was demonstrated to show similar results to the far-field condition results, based on MIL-STD-285 [16]. The details of the test setup are shown in Figure 3. The wide-band log-spiral antennas were used as transmitting and receiving antennas, connected with coaxial cables (RFOA1, RS-232), and delivered signals to a computer without loss. With distances of 20 mm from the concrete specimen to the testing apparatus, two antennas were installed perpendicular to the center of the specimen. Before SE measurement, calibrations were carried out to ensure correct measurement from the nominal value on the receiver. Then, the test specimen was placed between two antennas, and values were measured. The EMI shielding test was repeated five to six times for each specimen, and the average values were used in this research.

### 2.3. Detail Setup of Impact Test

For the evaluation of impact resistance, a drop-weight impact test was conducted on all specimens under low-velocity impact loading conditions, employing a user-defined setup. The detailed setup of the drop-weight impact test is shown in Figure 4. The clear span of the test specimen was 250 mm, and all sides of the specimen were fixed to prevent rebounding upon impact. The cylinder frustum head (a diameter of 70 mm) was used as the drop-weight to prevent severe damage to the test specimen. The cylinder frustum head was dropped on the center of the test specimen, applying a weight of 100 kg from a height of 200 mm (increased at each loading step by 100 mm). Two load cells were mounted on either side of the supports to measure the reaction force. The maximum and residual deflections were measured using a laser LVDT (KL4-120NV, Tokyo, Japan), which was placed at the center of the bottom side of the test specimen. All instruments were connected to a dynamic data logger (DEWE-43V, Trbovlje, Slovenia), which measured 200,000 data counts per second.

## 3. Experimental Results and Discussion

### 3.1. Shielding Effectiveness of HSDC-Strengthened Specimen

HSDC consisting of polyethylene and steel fibers with no specific orientation was developed and investigated. Well-known for their strength-improving qualities, hybrid fibers help to reduce cracking and tensile strain, while simultaneously enhancing electrical conductivity and improving toughness [13,14,15].

The results of the EMI SE test for HDSC of differing thicknesses are shown in Figure 5, comparing it with concrete reinforced with two different volumes of steel fiber (test results from Yuan et al., 2021) [5,6]. Briefly, 0.75 and 1.50 vol.% of steel fiber contents in reinforced concrete specimens show similar EMI SE behavior. Specimens reinforced by 0.75 vol.% of steel fibers were effective in surpassing the percolation threshold. According to previous research [2,5,6], when steel fibers are randomly dispersed through concrete, a matrix is formed that produces a conductive grid through an effective and continuous electrical pathway, thus increasing the EMI SE of the specimen. Therefore, the concrete specimens that were reinforced with steel fibers showed an SE of 40 dB to 50 dB, in a range of 500–1400 MHz, respectively, without resonance. Thicknesses of 100 and 200 mm in HDSC reinforcement demonstrate a continuation in this SE growth, between the ranges of 400 and 1000 MHz. Additionally, the SE of the specimen was 40 dB to 50 dB when between 1000 and 1400 MHz. It can be surmised that the SE performance of fiber-reinforced concretes (F0.75-N and F1.50-N) were higher than that of HSDC, even though HSDC presented greater fiber contents (than that of F0.75-N specimens), comparatively speaking. This performance is influenced by the fiber lengths used, with HSDC presenting a length of *l_f_* = 19.5 mm, and reinforced concrete showing a length of *l_f_* = 35 mm, thus leading to the conclusion that the fibers are significantly shorter in HSDC. Fiber length, size, shape, and aspect ratio all played an important role in the SE of the concrete when reinforced with fibers [17].

The results for the EMI SE tests of concrete specimens with varying thicknesses of HSDC are shown in Figure 6. As a result of increasing HSDC thickness, SE displayed an increase at the same test frequencies, additionally seeing an improved result in increasing frequencies; this relationship is observed in Figure 6a. One-layer 5 mm-thick HSDC concrete strengthening saw a greater SE response than that of 200 mm-thick standard non-strengthened concrete, and like that of 300 mm-thick concrete. SE was noticeably seen to increase when double-layer HSDC strengthening was introduced, as shown in Figure 6b. Although the double-layer reinforcement improves shielding effectiveness over single-layer reinforcement, the shielding effectiveness is increased at a certain frequency owing to the resonance phenomenon in the double-layer reinforcement. An increasing HSDC thickness also leads to the presence of resonance peaks that shift toward lower test frequencies. The cancellation of reflected waves in the first and second HSDC layers explains this occurrence. In the scenario where the matrix and absorber materials experienced interference resulting from the reflected waves, the distance of strengthened layers was approximately one-quarter of the propagating wavelength, multiplied by an odd number (thickness (w) = (odd number * propagating wavelength of selected materials)/4) [2,18]. Furthermore, specimens strengthened with the same HSDC parameters but altering the initial conditions showed similar SE results; this occurred over the entire frequency range (see Figure 6c,d).

Additionally, it is evident that the increase in spacing of two strengthening layers in HSDC does not directly affect growth in SE for the reinforced specimen, therefore an increase in SE is not simply achieved through the application of increased thickness in HSDC, as shown in Figure 7a. Double-layer strengthening of the same thickness (total thickness) showed results that were slightly higher than single-layer occurrences, with results showing 1.5–9.1 dB in the range of 600–1400 MHz. It can be speculated based on these results that there is a direct correlation between both spacing and thickness of HSDC reinforcement as a key factor in the increased strength of SE. An e-exponential function best describes the SE properties of concrete when compared with the thickness of strengthening material. This e-exponential can be best predicted with the use of the following model that has been proposed through research (experimentation):(1)$$SE_{HSDC}=SE_{plain}+e^{(a\left(\ln\left(1+T\right){)}^{b}\right)}$$
where $SE_{plain}$ denotes specimens without strengthening, *a* and *b* are the regression coefficients, and *T* is the thickness of HSDC.

Experimental results showed a strong agreement with that of predicted values in specimens strengthened by single- and double-layer HSDC concrete, showing coefficients of determinations (R^2^) of 0.989 and 0.944, respectively. An improvement of 35.4–65.7% in SE properties is observed in specimens strengthened by double-layer HSDC compared to that of single-layer strengthening when referring to the proposed prediction equation. Interestingly, HSDC strengthened members for both single- and double-layer strengthening saw the majority change in frequencies between 1000–1400 MHz.

### 3.2. Shielding Effectiveness of Impact-Damaged Specimen

#### 3.2.1. The Results of Drop Weight Impact Test

Figure 8 and Figure 9 represent the number of drop-blows versus the maximum deflection, reaction force, crack numbers, and crack width of specimens tested under impact conditions. The reaction force sees a sharp and sudden decrease after a certain count of drop blows (varies from member to member), indicating failure in the specimen. The calculation of maximum reaction force, obtained through the measuring of blows, gave an insight into the improved capacity when subjected to the strengthening of either HSDC.

Figure 8 shows the impact test results of specimens strengthened by single-layer HSDC. The specimen with 5 mm single-layer HSDC exhibited Max. reaction force after first drop weight and failure at second. Specimens strengthened by 10- and 20-mm thicknesses failed at the fourth and fifth loading steps. However, all specimens strengthened by single-layer HSDC exhibited the lowest properties (such as reaction force, deflection, cracking resistance) compared with specimens strengthened with double-layers. This is due to the bottom side of the specimen with no strengthening, thus large cracks and/or scabbing was exhibited at each loading step. Remarkably, the impact resistance properties of specimens strengthened by single-layer HSDC displayed slight increases in thickness.

The thickness of HSDC was seen to have a significant influence on the deflection and reaction force of members reinforced with HSDC; this can be seen in Figure 9. A more gradual increase in reaction force during the later stages of impact testing was seen as the thickness of HSDC increased; more specifically, H20D shows similar reaction forces to that of HD100 before the fourth impact. The total reaction forces were 2.2 and 1.6 times bigger than that of specimens H5D and H10D, respectively, in the case of specimen H20D. Maximum displacement showed a reduction as the HSDC thickness increased; this was particularly noticeable in the case of H20D. Steel fiber improves impact resistance by reducing the maximum and residual displacements and increasing the energy dissipation capacity; the crack propagation velocity is greater than the transfer velocity and shorter cracks are obtained at higher loading rates.

Specimens under the first impact load were shown to present radial and radioactive cracking, with crack widths occurring from 0.05 to 0.15 m. The completion of the second impact load brought forward the evidence of diagonal fractures, more specifically in the specimens strengthened by 5 or 10 mm of HSDC (H5D, H10D). The fourth and fifth drop blows saw crack widths of 5.00 mm and 5.50 mm in H5D and H10D, respectively. Unsurprisingly, the H20D sample that had the largest HSDC thickness was the most resilient when considering impact loading. Furthermore, the cracking patterns observed on the bottom side of the test specimen demonstrated hairline crack widths as the primary form of cracking, with a width of approximately 0.05 mm after the first impact. Crack widths of 0.70 and 1.10 mm were measured after the fourth and fifth blow, respectively, and were seen to be the maximum crack widths at that time. In comparison to the H5D and H10D samples, the strengthening materials in H20D were more successful in controlling and mitigating crack growth. The sixth impact load resulted in new deformation to the specimens on their underside, showing fibers bridging the pre-existing large diagonal cracks; additionally, wide diagonal cracks presented, originating in the center of the specimen, and ending at its edges. Large diagonal cracks resulting from fracturing are reproduced in Figure 10.

Furthermore, the influence of the number of strengthening layers and thickness on the impact capacities of specimens, using total imparted energy, is shown in Figure 11. The total imparted energy increased with strengthening thickness, no matter the strengthened specimen, whether it was single- or double-layered. In contrast, the double-layer sample strengthened with HSDC resulted in a significant increase of impact capacities, with the total energy imparted to the H20D specimen being more than five times and 3.5% larger than that imparted to the H5S and H5D specimens, respectively. The increased impact resistance properties of the specimens correlated with the strengthening side numbers and thickness; similar strength trends were obtained for specimens at 20% strengthening area ratios. This was presumably because the strengthening thickness of the specimen increased fiber contents; thus, as determined in previous research, concrete reinforced with fibers exhibited smaller displacement amplitudes under impact loading and was able to undergo larger displacements and reaction force amplitudes before failure [11,19].

#### 3.2.2. Shielding Effectiveness after Impact Tests of Strengthened Specimens

As found in previous research [6,20], the SE of the concrete specimen is mainly influenced by effective thickness variation, which is effective against external loads or environmental factors. Thus, the SE capability was evaluated for the specimens after low-velocity drop-weight tests in this study, using the SE decrease ratio versus the damage area ratio (which included the crack and scabbing area), as shown in Figure 12.

Specimens strengthened with 20 mm of HSDC exhibited the lowest SE decrease ratio compared to those of the other specimens, which exhibited as approximately 7.1 and 24.7%, respectively. This is because specimens strengthened by 5 and 10 mm of HSDC exhibited large amounts of spalling at the center of the bottom surface in the impact region, which significantly decreased the effective thickness. However, specimens strengthened with a single layer exhibited smaller values of SE decrease ratios compared with double-layer-strengthened specimens, at similar damage area ratios. This is presumably because the single-layer strengthening method confirmed slightly improved SE values; thus, comparatively small values were measured before and after the impact test. The specimens strengthened by a double layer exhibited higher SE levels in terms of damage compared to single-layer-strengthened specimens (with and without an impact test). Furthermore, although various areas of damage decreased the effective thickness of test specimens, the SE decrease ratio exhibited similar values at similar damage area ratios in those specimens without different strengthening thickness, according to the uniformly distributed steel fiber.

## 4. Conclusions

This research was conducted to assess the SE and impact resistance properties of concrete walls strengthened with high-strength high-ductility concrete. A variety of strengthening methods were adopted to evaluate the SE and impact resistance. From this investigation, the following conclusions can be drawn:The specimens strengthened with HSDC displayed an increase in SE with increasing strengthening thickness, which was 2.4–35.3, 7.0–31.9, and 6.9–43.8 dB, respectively, comparatively higher than specimens without strengthening. For the single- and double-layer HSDC strengthening, a strengthening area ratio greater than 40% produced an SE that was over 20 dB.The properties of specimens reinforced with a double layer were greater than those of specimens reinforced with a single layer (such as reaction force, deflection, and cracking resistance). The specimen’s improved impact resistance characteristics corresponded with the strengthening side numbers and thickness, and similar strength trends were achieved for specimens with 20% strengthening area ratios.Specimens reinforced by 20 mm of HSDC had the lowest SE decrease ratio when compared to other specimens reinforced by 5 and 10 mm of HSDC, which showed around 7.1 and 24.7%, respectively. This is due to large amounts of spalling in the center of the bottom surface in the impact region in specimens strengthened by 5 and 10 mm, which greatly reduced the effective thickness. Although multiple incidents of damage were displayed and lowered the effective thickness of test specimens, the SE reduction ratio demonstrated equal values in similar damage area ratios of specimens that had uniformly dispersed steel fiber.

Therefore, concrete structures that have been strengthened by HSDC could be chosen in applications exhibiting the need for increased SE.

## Figures and Tables

**Figure 1 materials-14-07773-f001:**
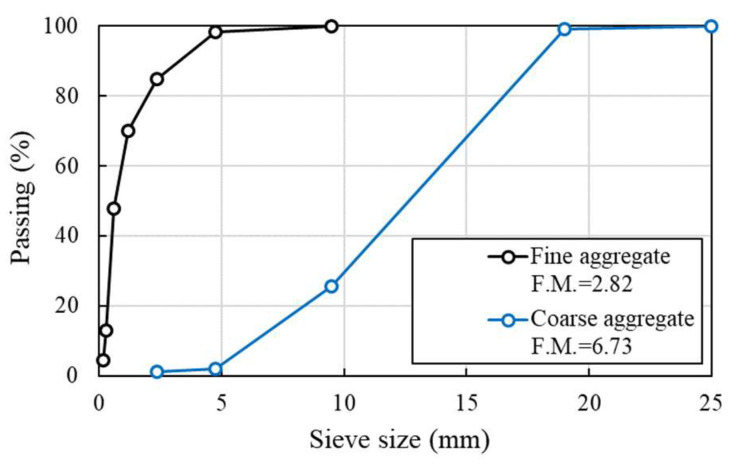
Sieve analysis of aggregates.

**Figure 2 materials-14-07773-f002:**
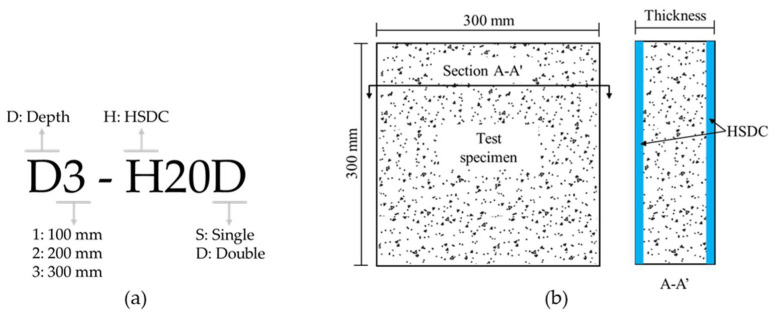
Details of the test specimen. (**a**) Designation of test specimens; (**b**) Detail of the test specimen.

**Figure 3 materials-14-07773-f003:**
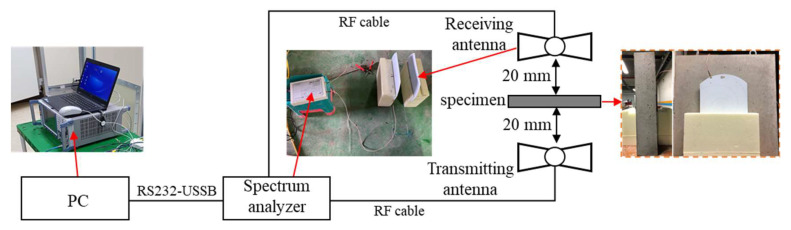
Test setup of shielding effectiveness.

**Figure 4 materials-14-07773-f004:**
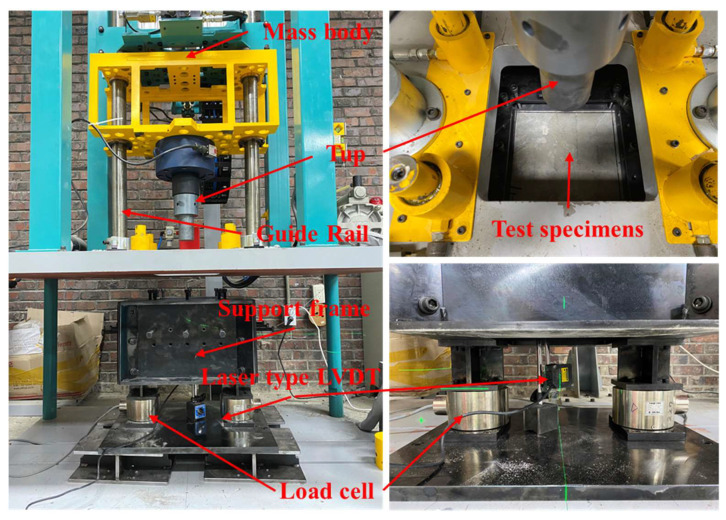
Details of the drop weight impact test.

**Figure 5 materials-14-07773-f005:**
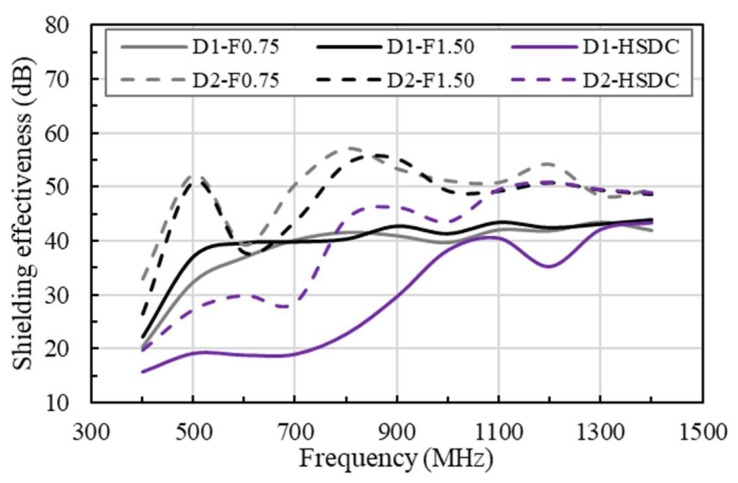
Comparison of shielding effectiveness in concrete reinforced with varying volumes of fibers.

**Figure 6 materials-14-07773-f006:**
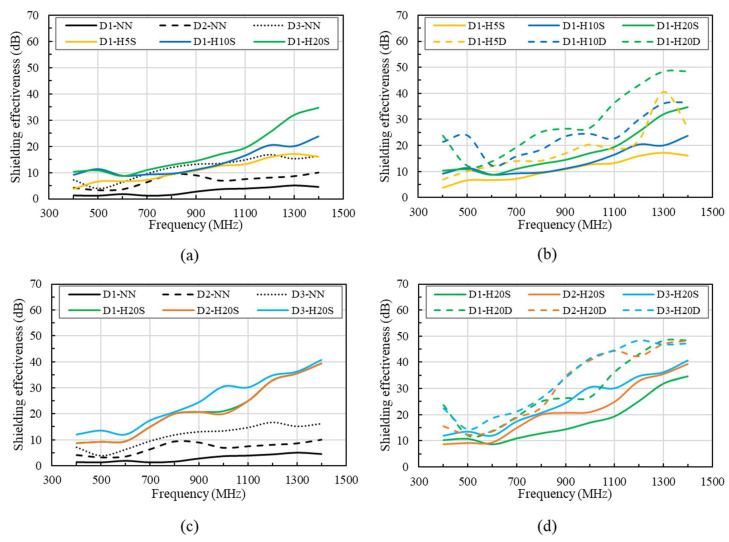
Comparison of shielding effectiveness for differing HSDC combinations: (**a**) single layer, 100 mm thickness specimens; (**b**) double layer, 100 mm thickness specimens; (**c**) single layer, different thickness specimens; (**d**) double layer, different thickness specimens.

**Figure 7 materials-14-07773-f007:**
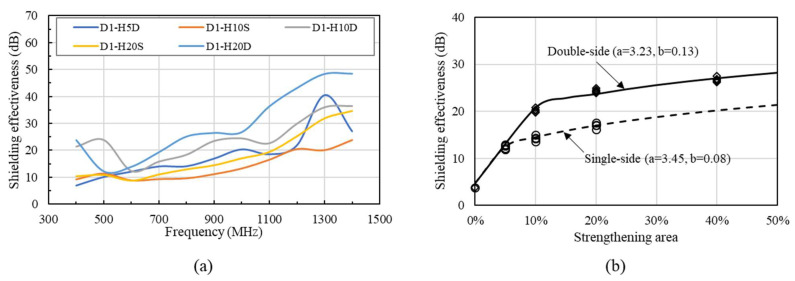
Comparison of shielding effectiveness of single- and double-layer strengthening methods: (**a**) Different HSDC thicknesses, (**b**) comparison of varying strengthening areas.

**Figure 8 materials-14-07773-f008:**
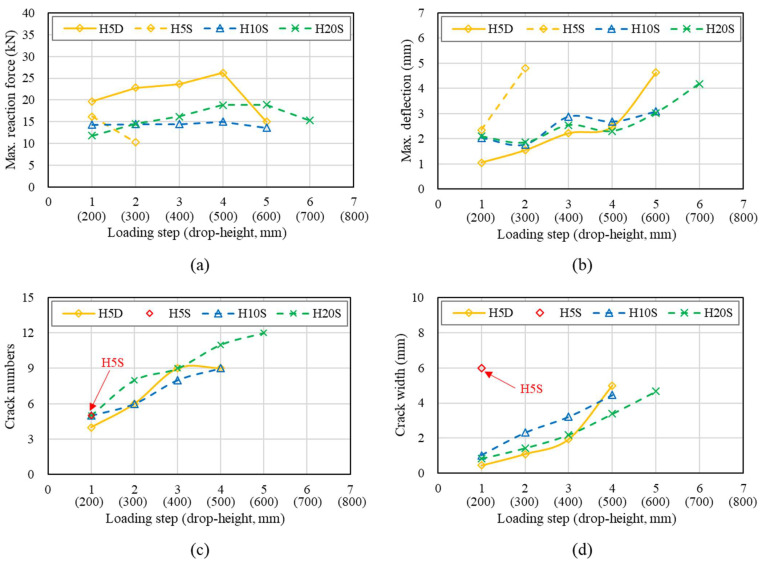
Impact test results of single-layer HSDC-strengthened specimens: (**a**) maximum reaction force at each load; (**b**) maximum deflection at each load; (**c**) number of cracks at each load; (**d**) maximum crack width at each load.

**Figure 9 materials-14-07773-f009:**
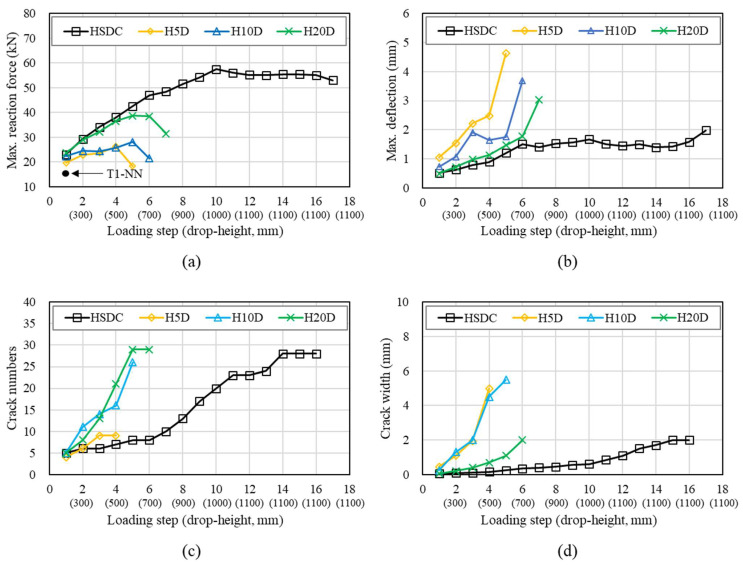
Impact test results of double-layer HSDC-strengthened specimens: (**a**) maximum reaction force at each load increment; (**b**) maximum deflection at each load increment; (**c**) crack numbers at each load increment; (**d**) maximum crack width at each load increment.

**Figure 10 materials-14-07773-f010:**
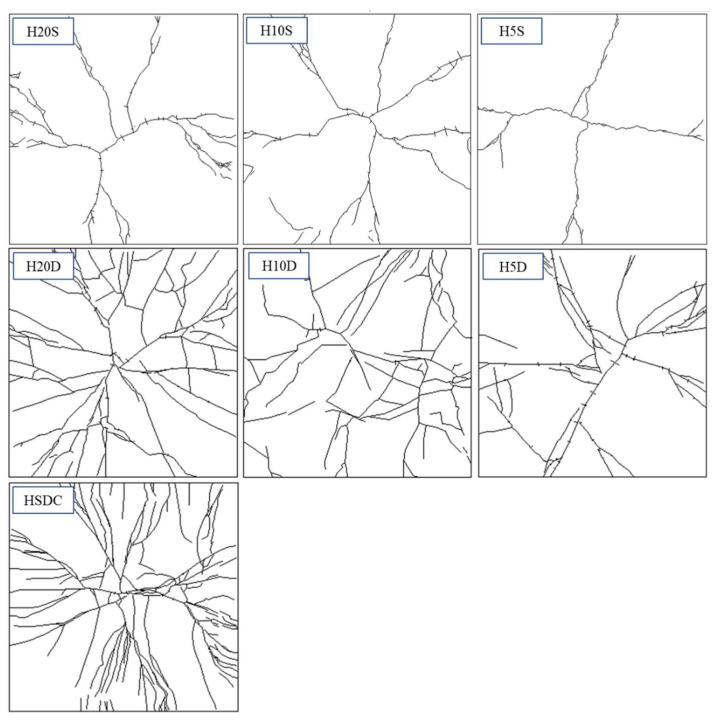
Failure mode after the final impact test.

**Figure 11 materials-14-07773-f011:**
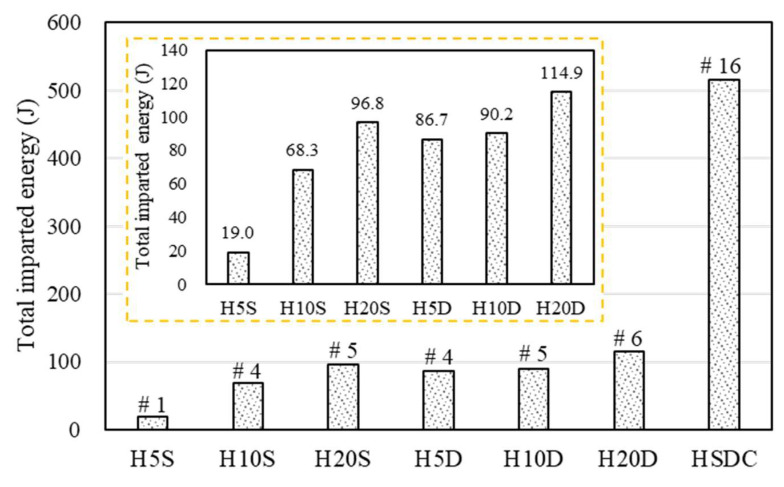
Total imparted energy until failure.

**Figure 12 materials-14-07773-f012:**
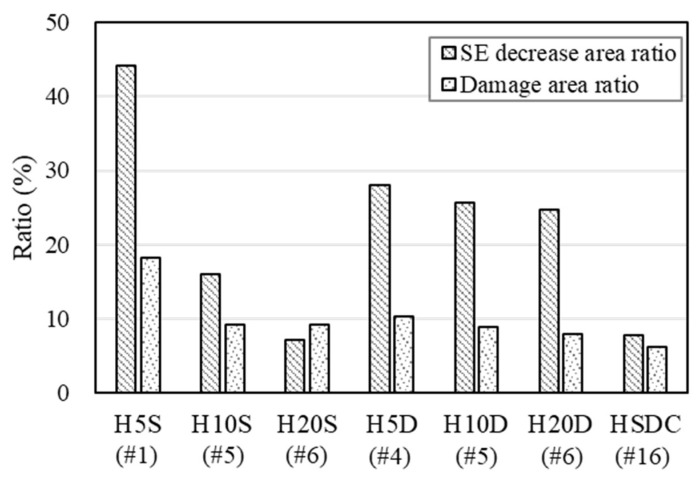
Comparison of shielding effectiveness decrease ratio and damage area ratios.

**Table 1 materials-14-07773-t001:** Mix proportions (by weight).

	W/C	W	C	Fine Aggregate	Coase Aggregate	Silica Fume	Silica Filler	Silica Sand	Steel Fiber	Polyethylene Fiber	SP
NC	0.43	0.43	1.00	2.15	2.42	-	-	-	-	-	0.8%
HSDC	0.172	0.215	1.00	-	-	0.25	0.30	1.10	1.0%	0.5%	3.0%

NC = normal concrete; HSDC = high-strength high-ductility concrete; W/C: water-to-cement ratio; W: water; C: cement; SP: superplasticizer.

**Table 2 materials-14-07773-t002:** Chemical properties of materials.

	Surface Area (cm^2^/g)	Density (g/cm^3^)	Chemical Composition (%)						
			SiO_2_	Al_2_O_3_	Fe_2_O_3_	CaO	MgO	SO_3_	Na_2_O
Cement	3492	3.15	21.2	4.7	3.1	62.8	2.8	2.1	-
Silica fume	200,000	2.20	96.0	0.3	0.1	0.4	0.1	<0.2	-
Silica sand	2990	2.7	99.7	0.14	0.016	0.01	0.01	-	0.01

**Table 3 materials-14-07773-t003:** Properties of steel fiber.

	Diameter, *d_f_* (mm)	Length, *l_f_* (mm)	Aspect Ratio (*l_f_*/*d_f_)*	Density (g/cm^3^)	Tensile Strength (MPa)	Elastic Modulus (GPa)
Steel fiber	0.2	19.5	97.5	7.8	2650	200

**Table 4 materials-14-07773-t004:** Strength results (after 28 days).

Variable	Compressive Strength (MPa/CV)	Flexural Strength (MPa/CV)	Tensile Strength (MPa/CV)	Remarks
NC	44.6/0.2	25.7/0.4	3.9/0.5	Splitting tensile strength test
HSDC	122.3/0.1	22.9/6.5	9.7/1.6	Direct tensile strength test

CV = coefficient of variation.

**Table 5 materials-14-07773-t005:** Details of specimens.

	Variable	Thickness	HSDC	
			Thickness	Layer
1	D1-HSDC	100 mm	-	-
2	D2-HSDC	200 mm	-	-
3	D1-H5S	100 mm	5 mm	single
4	D1-H5D	100 mm	5 mm	double
5	D1-H10S	100 mm	10 mm	single
6	D1-H10D	100 mm	10 mm	double
7	D1-H20S	100 mm	20 mm	single
8	D1-H20D	100 mm	20 mm	double
9	D2-H20S	200 mm	20 mm	single
10	D2-H20D	200 mm	20 mm	double
11	D3-H20S	300 mm	20 mm	single
12	D3-H20D	300 mm	20 mm	double

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
