# Peer review of "Shielding Effectiveness and Impact Resistance of Concrete Walls Strengthened by High-Strength High-Ductility Concrete"

_materials, 2021, doi:10.3390/ma14247773_

Round 1

Reviewer 1 Report

Changes must be applied to this manuscript to make it suitable for the readership of the journal. Please find below some of the recommendations which may help improve the quality of the paper:

  1. The manuscript contains many grammatical, typographical and spelling errors apart from numerous incomplete sentences. The authors should have the paper thoroughly English proofed.
  2. Abstract lacks motivation of the study. The authors need to rewrite the abstract. Highlight the scientific value added by your paper in your abstract. The problem should be stated and summarized. The abstract should clearly describe the core of the problem you are addressing, what you did, found, and recommended to the readers. It will help prospective readers of the abstract to decide if they wish to read the entire article.
  3. Too many abbreviations make it difficult for readers to understand this article. As a reader, I would like to see the abbreviations listed in a table after the abstract. I must repeat my comment on the use of abbreviations throughout the manuscript. You are the authors of this paper; you have comprehensive knowledge about the terminologies used in this paper. But we, the readers of this paper, have not. There are many uncommon abbreviations in the manuscript, which decreases the paper's readability and understandability. I recommend that the authors employ the open form for some abbreviations. Please put yourself into readers' shoes, and try to simplify the language of the paper.
  4. Authors need to add the significance section of this study regarding past studies to highlight this study's novelty. What is the novelty of this work? Would you please discuss similar past studies or reports? What is new, and where is the science?
  5. The manuscript is editorially weak and needs a substantial review effort by authors to improve the manuscript. Authors themselves must carry out the editorial effort as it is not justified for reviewers to give detailed feedback on improvements desired unless these are small in number.
  6. The conclusions section should be improved. The conclusions drawn in this paper have already been pointed out in previous studies.

Author Response

Response to Reviewer 1 Comments

Point 1: The manuscript contains many grammatical, typographical, and spelling errors apart from numerous incomplete sentences. The authors should have the paper thoroughly English proofed.

Response 1: Thank you for your valuable comments. The errors, as you stated, have been corrected.

Point 2: Abstract lacks motivation of the study. The authors need to rewrite the abstract. Highlight the scientific value added by your paper in your abstract. The problem should be stated and summarized. The abstract should clearly describe the core of the problem you are addressing, what you did, found, and recommended to the readers. It will help prospective readers of the abstract to decide if they wish to read the entire article.

Response 2: Thank you for your comments. The abstract has been revised based on your comments. I really appreciate your comment. I rewrote the abstract “Following the 4th industrial revolution, electronic and data-based technology is developed. However, research on enhancing electromagnetic interference (EMI) shielding and physical protection performance of structures incorporating these technologies are insufficient. Therefore, in this study, for the study for the improvement of EMI shielding and structural performance of the structure, twelve concrete walls were fabricated and tested to determine shielding effectiveness and drop-weight impact resistance. Concrete walls strengthened by three thickness types of high strength high ductility concrete (HSDC) were considered. The test results showed that the shielding effectiveness with strengthening thickness increased, which were approximately 35.6-46.2%. Specimen strengthened by more than 40% and 10% of strengthening area ratio of single- and double-layer, respectively, exhibit more than 20 dB shielding effectiveness. Moreover, the relationship between damaged area ratio and shielding effectiveness was evaluated according to the drop-weight impact test. The structural performance and EMI shielding effectiveness improved as HSDC thickness increased.”.

Point 3: Too many abbreviations make it difficult for readers to understand this article. As a reader, I would like to see the abbreviations listed in a table after the abstract. I must repeat my comment on the use of abbreviations throughout the manuscript. You are the authors of this paper; you have comprehensive knowledge about the terminologies used in this paper. But we, the readers of this paper, have not. There are many uncommon abbreviations in the manuscript, which decreases the paper's readability and understandability. I recommend that the authors employ the open form for some abbreviations. Please put yourself in the readers' shoes and try to simplify the language of the paper.

Response 3: Thank you for your kind comments. To avoid the readability of this paper from being weakened by too many abbreviations, the names of the experimental variables have been explained in Figure 2 (a).

Point 4: Authors need to add the significance section of this study regarding past studies to highlight this study's novelty. What is the novelty of this work? Would you please discuss similar past studies or reports? What is new, and where is the science?

Response 4: Thank you for your valuable comments on the development of this paper. The shielding effectiveness of the undamaged structure was evaluated in the previous studies(Choi et al., 2020; Jung et al., 2019; Quintana et al., 2018). However, research on residual shielding effectiveness after structural damage is insufficient. Therefore, this paper high-strength high-ductility concrete was used to improve shielding effectiveness and structural performance and evaluated the residual shielding effectiveness due to damage to the structure after the impact test. As your comments, I rewrote “EMI shielding is generally performed by heavy metal plates, conductive paints/spraying, or conductive mesh, which require high initial costs and continuous maintenance even if they are effective. Few studies have conducted shielding- and structural performance of undamaged structures. [1-6]. Thus, it is necessary that to develop widely used construction materials such as conductive concrete or reinforcing method with a metal grid to ensure high shielding efficiency compared to cost and evaluate shielding effectiveness of damaged structures due to external load.” introduction.

Choi, J., Yuan, T., Hong, S., & Yoon, Y. (2020). Evaluating of Electromagnetic Shielding Characteristics of Reinforced Concrete Using Reinforcing Details. Journal of the Korean Society of Hazard Mitigation, 20(5), 245-254.

Jung, M., Lee, Y.-S., & Hong, S.-G. (2019). Effect of incident area size on estimation of EMI shielding effectiveness for ultra-high performance concrete with carbon nanotubes. IEEE Access, 7, 183105-183117.

Quintana, S., de Blas, J., Peña, J., Blanco, J., García, L., & Pastor, J. (2018). Design and operation of a real-scale electromagnetic shielding evaluation system for reinforced composite construction materials. Journal of Materials in Civil Engineering, 30(8), 04018162.

Point 5: The manuscript is editorially weak and needs a substantial review effort by authors to improve the manuscript. Authors themselves must carry out the editorial effort as it is not justified for reviewers to give detailed feedback on improvements desired unless these are small in number.

Response 5: Thank you for taking your valuable time to comment, edited parts lacking in this paper have been corrected and reflected in the manuscript. I thank you for your efforts in the development of this paper.

Point 6: The conclusions section should be improved. The conclusions drawn in this paper have already been pointed out in previous studies.

Response 6: I am grateful for the comments on the development of this paper. I have revised the conclusion as you said.

Reviewer 2 Report

See in the attachment

Author Response

Response to Reviewer 2 Comments

Point 1: The abstract can be enhanced, there is need to add a brief introduction/background as the opening statement of the abstract

Response 1: Thank you for your valuable comments. The abstract had rewritten as you said. I really appreciated your effort to develop this paper.

Point 2: Line 37-38: The statement requires reference.

Response 2: Thank you for your valuable comments. References are written after the sentence you commented on. Thank you again.

Point 3: Line 38-41: The statement requires reference as it is missing.

Response 3: Thank you for your valuable comments. References are added following the sentence on which you commented. Thank you one again.

Point 4: Line 42-44: Reference is required.

Response 4: Thank you for your valuable comments. Following the sentence on which you commented, references are added. Thank you one again.

Point 5: Authors stated that “Many studies have been conducted on the improvement of shielding performance by a reinforcement with metallic grid and mesh, which are used as structural reinforcements instead of reinforcing bar”, but they haven’t reviewed any of the studies and incorporated into their manuscript. I suggest you review some latest work on the shielding improvement and add to the introduction section.

Response 5: Thank you for your valuable comments. Based on your comments, references related to the sentence have been added. Thank you for your effort to improve this paper.

Point 6: In the methodology section, there is need to first introduce the          materials before the mix design or methodology.

Response 6: Thank you for your valuable comments. As you said, I have added description about the materials description in the introduction.

Point 7: Line 87-88: Which Standard or method was adopted for the mix design of the conventional concrete.

Response 7: Thank you for your valuable comments before replying. ACI 318-19 was referenced for the concrete mix design conducted in this paper.

Point 8: There is need to add the particle size gradation of the aggregates                                 used.

Response 8: Thanks for your comments. The gradation of particle size of the aggregate is added in figure 1.

Point 9: Table 1 and Table 2 can be merged.

Response 9: Thank you for your valuable comments. Table 1 and 2 are merged.

Point 10 The properties of the cement, silica fume should be added in  Tabular form (Chemical properties)

Response 10: The chemical properties of cement and silica fume have been prepared and added in a table. Thanks for your comments.

Point 11: The properties of the silica sand and steel fibre can be added also.

Response 11: Than The chemical and physical properties of silica sand and steel fiber have been prepared and added in a table. Thanks for your comments.

Point 12: The result was well presented; the only problem is that authors  did not compare their findings with previous studies.

Response 12: Thank you for your valuable comments before replying. Compared with previous studies, this study reinforced the structure using HSDC and evaluated the structure shielding and structural performance through shielding and impact tests. Afterward, the residual shielding performance of the damaged structure was evaluated to show the difference form the previous research. In addition, research results on the improvement of shielding and structural performance according to mixed materials are written in the introduction of the results section. I thank you for your efforts and interest in the development of this paper.

Point 13: Line 58-59, this should not be part of result discussion but more  of a recommendation.

Response 13: Thank you for your valuable comments. This sentence has been deleted from this paragraph. Thank you again.

Point 14: Line 181-196, authors only reported their findings but did not give any reasons for why the obtained such results.

Response 2: Thank you for your valuable comments before replying. It was written with additional descriptions of the shielding test results.

Point 15: Fig 10 and Fig 11, there is need for error bars to be added. Most of the results findings were not backed up with reasons of their occurrence, it is more or less like the results were only reported but not well discussed.

Response 15: Thank you for your kind words. It is difficult to draw an error bar because the experimental results were produced one for each variable. Thank you for your attentive review.

Point 16: I suggest the results to be discussed more appropriately.

Response 16: As it was judged that the contents of the impact test result part were insufficient, the contents were added as ‘Steel fiber improves impact resistance by reducing maximum and residual displacements and increasing energy dissipation capacity because crack propagation velocity is greater than transfer velocity and shorter cracks are obtained at high loading rates.’. Thank you for taking your valuable time to comment.

Point 17: The conclusions are adequate.

Response 17: Thanks for the comments to improve this paper. The results have been corrected.

Reviewer 3 Report

The authors presented an experimental investigation on the performance of concrete strengthened by one or two layers of different thicknesses of  high strength high-ductility concrete under falling impact load and their shielding effectiveness.

The paper is well-structured and well written with a well explained research significance and recent literature review.

The paper could be accepted after minor revision.

  1. The need for a hybrid fiber reinforcement (steel and polyethylene fibers) should be elucidated
  2. In the introduction section, there is a need to explain why the impact tests are needed.
  3. How the authors ensure that the surface preparation of the specimens before strengthening was effective?
  4. Figure 3 is not very clear and probably a schematic figure showing the whole set up including the specimen under test is needed.

Author Response

Response to Reviewer 3 Comments

Point 1: The need for a hybrid fiber reinforcement (steel and polyethylene fibers) should be elucidated.

Response 1: Thanks for your kind words. PE fiber reduces micro-cracks and steel fiber reduces macro-cracks. Details of the study results are as follows ‘Enhancing the tensile capacity of no-slump high strength high ductility concrete’. Thank you again.

Point 2: In the introduction section, there is a need to explain why the impact tests are needed.

Response 2: Thank you for your valuable comments in advance of the reply. As additionally written in the introduction of this paper, the previous research evaluated the shielding performance of undamaged structures, but studies on the evaluation of residual shielding performance of damaged structures are lacking. Therefore, in this study, an impact test was conducted to induce damage to the structure, and the shielding and structural performance according to HSDC reinforcement was evaluated.

Point 3: How the authors ensure that the surface preparation of the specimens before strengthening was effective?

Response 3: In this study, the order of making the specimens is as follows. After pouring HSDC according to the set thickness, normal concrete was poured on it to produce specimens. The above manufacturing method was manufactured by referring to the existing literature ‘Evaluation of Electromagnetic Shielding and Impact Resistance Effectiveness for Metallic Grid Strengthening Concrete Walls’. Thank you for your effort to develop this paper.

Point 4: Figure 3 is not very clear and probably a schematic figure showing the whole set up including the specimen under test is needed.

Response 4: Thank you for your interest and comments on the development of this paper. The figure has been modified and added to better understand the setup of the impact test. Thank you again.